# Charge-Density Waves vs. Superconductivity: Some Results and Future Perspectives

**Giulia Venditti** [1,*] and **Sergio Caprara** [2]

1 SPIN-CNR Institute for Superconducting and Other Innovative Materials and Devices, Area della Ricerca di Tor Vergata, Via del Fosso del Cavaliere 100, 00133 Rome, Italy
2 ISC-CNR and Department of Physics, Sapienza University of Rome, P.le A. Moro 5, 00185 Rome, Italy; sergio.caprara@roma1.infn.it or sergio.caprara@uniroma1.it
* Correspondence: giulia.venditti@spin.cnr.it

**Abstract:** Increasing experimental evidence suggests the occurrence of filamentary superconductivity in different (quasi) two-dimensional physical systems. In this piece of work, we discuss the proposal that under certain circumstances, this occurrence may be related to the competition with a phase characterized by charge ordering in the form of charge-density waves. We provide a brief summary of experimental evidence supporting our argument in two paradigmatic classes of materials, namely transition metal dichalcogenides and cuprates superconductors. We present a simple Ginzburg–Landau two-order-parameters model as a starting point to address the study of such competition. We finally discuss the outcomes of a more sophisticated model, already presented in the literature and encoding the presence of impurities, and how it can be further improved in order to really address the interplay between charge-density waves and superconductivity and the possible occurrence of filamentary superconductivity at the domain walls between different charge-ordered regions.

**Keywords:** superconductivity; charge-density wave; competition

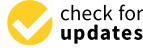



## 1. Introduction

The physics of (quasi) two-dimensional superconductors is attracting increasing attention thanks to the recent advances in the synthesis and tailoring of two-dimensional electron systems. Among these, we mention oxide interfaces [1], field-effect-controlled devices [2], atomic monolayers grown by means of molecular beam epitaxy [3], and thin flakes obtained by exfoliation [4]. Nowadays, it is even possible to single out and combine two-dimensional materials, such as graphene, individual layers of transition metal dichalcogenides and high-critical-temperature superconducting cuprates, paving the way toward the engineering of functionalized two-dimensional systems [5].

All the above systems share the common feature of being highly crystalline and therefore represent the ideal context to study intriguing phenomena [6], such as, e.g., Ising or topological superconductivity, bypassing or at least reducing the obnoxious effect of defect-induced disorder.

One of the noticeable properties of many two-dimensional systems is the occurrence of a low-temperature residual metallic state, which has been proposed as a new quantum state of matter, i.e., the so-called "quantum metal" [7] or "anomalous metal" [8], which escapes the customary fate of two-dimensional electron systems, which are either Anderson insulating or superconducting at low temperatures. The residual metallic phase is reached upon lowering the temperature from the high-temperature metallic state and passing through an intermediate regime where the resistance is significantly suppressed, which is seemingly due to incipient superconductivity. Superconductivity is, however, unable to fully develop, e.g., due to the presence of a magnetic field or to other mechanisms at play. It is worth noting that this behavior has been observed also in highly crystalline two-dimensional systems, such as transition metal dichalcogenides or ultrathin ZrNCl films [9].

Some features of the resistivity vs. temperature curves $\rho(T)$ are characteristic of this peculiar state. Indeed, even if the conditions for a full development of superconductivity are met, e.g., because the external magnetic field is not strong enough, and the zero-resistance state is finally reached, the metal-to-superconductor transition is so broad that its width cannot be attributed to standard fluctuating phenomena, such as those described by the Aslamazov–Larkin or Halperin–Nelson theory, with any reasonable choice of the phenomenological parameters [10]. Moreover, the $\rho(T)$ curves display a rather pronounced tail on their low-temperature side. The presence of tails in the $\rho(T)$ curves is customarily associated to vortex-driven dissipation [11]; however, those tails are observed also in the absence of a magnetic field, for instance when the low-temperature superconducting phase is weakened by a (gate-controlled) reduction of the carrier density. This very fact witnesses that the customary mechanism of vortex-driven dissipation cannot be held responsible for the pronounced tails in the $\rho(T)$ curves.

An effective and successful interpretation for these anomalous features is offered by the comparison with the case of the two-dimensional electron gas formed at oxide interfaces such as, e.g., $LaAlO_3/SrTiO_3$, where the $\rho(T)$ curves exhibit strikingly similar behavior. Despite the highly crystalline structure of $LaAlO_3/SrTiO_3$ interfaces, the occurrence of tails in their $\rho(T)$ curves has been successfully interpreted in terms of nanoscale electron inhomogeneity [12–15]. Further support to the statement that oxide interfaces and other two-dimensional electron systems are (intrinsically) inhomogeneous comes from the observation of a low-temperature Griffiths state when the metal-to-superconductor transition can be driven by a magnetic field [16,17]. The claim of inhomogeneity might apparently clash with the highly crystalline structure of these systems. It is the good mobility of the charges that is at seeming odds with the transport observations, weakening customary scattering arguments that can be raised in metals at low temperatures (crystal defects and impurities). Instead, inhomogeneity may have an intrinsic origin [18,19] due to some mechanism that endows the electrons with a tendency to segregate, at least on a nanoscopic scale, resulting in a landscape of tiny but significant electron density modulations that are apt to locally enhance or suppress superconductivity [20].

Strong inhomogeneities of the superconducting condensate can indeed appear as a filamentary cluster whose origin might arise from different (intrinsic and/or extrinsic) sources, one of which might be the competition with another ordered state of matter. In this piece of work, we want to explore the scenario where this competing state is characterized by charge ordering in the form of charge-density waves. This competing mechanism seems to be relevant to at least two classes of (quasi) two-dimensional systems, namely some transition metal dichalcogenides and high-critical-temperature superconducting cuprates.

We point out that hints of filamentary superconductivity have been reported in other classes of materials, among which we find iron-based superconductors [21–24], whose phase diagram is very similar to that of cuprates, with an antiferromagnetic phase at low doping and a superconducting dome at higher hole doping, a pseudogap region and a stripe phase. Nonetheless, no evidence of charge ordering was found so far in iron-based materials, although spin-density waves could be a good candidate to consider in order to explore a similar scenario also in these systems. As a matter of fact, the stripe phase in iron-based superconductors has already been linked to the presence of spin-density waves [25–28].

The structure of the present paper is the following. In Section 2.1, we discuss evidence for the competition between superconductivity and charge-density waves in transition metal dichalcogenides, while in Section 2.2 we discuss the case of high-critical-temperature superconducting cuprates. In Section 3, we introduce a simple Ginzburg–Landau model to describe the competition of the two phases. In Section 4, we discuss in some more detail the physics of this competition. Our concluding remarks, as well as some directions to improve our understanding of the competition between superconductivity and charge-density waves, are found in Section 5.

## 2. Experimental Evidence for the Competition between Superconductivity and Charge-Density Waves in Some Selected Systems

As we have suggested above, there are at least two classes of (quasi) two-dimensional materials where the competition between superconductivity and charge-density waves may play a relevant role, namely some transition metal dichalcogenides and high-critical-temperature superconducting cuprates. Experimental evidence for such a competition to occur in these two paradigmatic systems is discussed below in Sections 2.1 and 2.2, respectively.

### 2.1. Transition Metal Dichalcogenides

The observation of periodic oscillations of the magnetoresistance induced by the Little–Parks effect in 1T-TiSe$_2$ [29] shows that the onset of superconductivity is directly related to the spatial texturing of the amplitude and phase of the superconducting order parameter, corresponding to a two-dimensional superconducting matrix. The authors of Ref. [29] infer that such a superconducting matrix originates from a matrix of incommensurate charge-density-wave states embedded in the commensurate charge-density-wave states, and they argue that their results give evidence for spatially modulated electron states to be fundamental to the appearance of two-dimensional superconductivity in 1T-TiSe$_2$ (see Figure 1). Indeed, it is clear that the peculiar features of the magnetoresistance mirror the spatial fluctuations of the superconducting (Cooper) pairing. A reasonable explanation for these peculiarities is based on the Little–Parks effect, whereby Cooper pairs are constrained to move in loops, forcing pairings to have a local character and to occur in well-defined regions. The observed length scale is associated with Cooper pairs trapping magnetic flux quanta. The remarkable occurrence of such a superconducting matrix in a single crystal calls for a pre-existing matrix of inhomogeneous electron states to stabilize it. Fluctuations of an underlying charge or spin order parameter appear to be relevant to the occurrence of superconductivity in a variety of physical systems. The suppression of the charge-density-wave transition from $T_{CDW} = 170\,\text{K}$ to $T_{CDW} = 40\,\text{K}$, alongside with the appearance of superconductivity and marked non-Fermi-liquid behavior of the metallic phase, strongly suggest that the charge-density waves play a role in 1T-TiSe$_2$. The authors of Ref. [29] speculate that domain walls form a periodic matrix in which commensurate charge-density-wave domains with constant phases are embedded in an incommensurate charge-density-wave matrix.

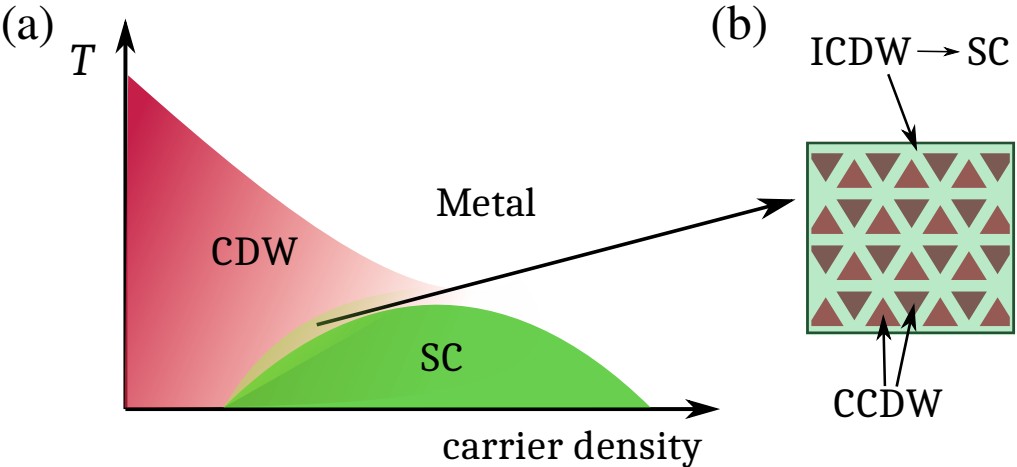

**Figure 1.** (**a**) Cartoon sketch of the phase diagram for transition metal dichalcogenides and (**b**) cartoon of the formation of a pattern of commensurate charge-density-wave regions (CCDW) embedded in an incommensurate charge-density-wave (ICDW) matrix, the latter possibly becoming superconducting.

Analogously, Little–Parks effect signatures have been reported also in $Li_x$-$TiSe_2$ [30], where magnetoresistance oscillations were observed in the superconducting phase along with an "anomalous metal" behavior. The transport characterization of electric field-controlled lithium intercalation in $TiSe_2$ provided evidence for the coexistence of commensurate and incommensurate charge-density waves, which is supported by the correlation between the charge-density wave and the magnetoresistance oscillations. Thus, also in this case, the authors speculate a periodic structure of commensurate charge-density waves encoded in an incommensurate charge-density-wave network, whereby the latter becomes superconducting at low temperatures. The average size $L$ of the commensurate regions was thus extrapolated in Ref. [30], as in Ref. [29], by comparing the magnetic field oscillations $\Delta B$ with the flux quantum, providing an opposite temperature and doping dependence of $L$ with respect to the case of ion-gated $TiSe_2$ presented in Ref. [29]. While in fact, in Ref. [29], $L$ quickly decreases with the carrier density, an increasing trend is observed in Ref. [30]. The other important difference between those two materials is their phase diagram. In [30], $T_{CDW}$ is in fact constant as a function of the carrier density, whereas instead, the intensity of the charge-density-wave response $I_{CDW}$ is suppressed with lithium intercalation.

The same trend of a constant $T_{CDW}$ and a decreasing $I_{CDW}$ is found in electric field-driven hydrogen intercalated $TiSe_2$, which is probed through transport and magneto-transport measurements, X-ray diffraction, Raman spectroscopy and nuclear magnetic resonance [31]. As in the previous case, the insensitivity of $T_{CDW}$ to increasing doping can be interpreted as a lack of full in-plane percolation of the dopants occurring on length scales smaller than $\sim 10\,\mu m$. On the other side, the charge-density-wave order might still appear in H-rich regions; two possible explanations could be either that the dopants suppresses long-range order and the charge-density waves survive only on short-range scales, or, similarly to the case of $Cu_x$-$TiSe_2$ [32], that the opening of the charge-density-wave gap is shifted below the Fermi level with an increasing density of charge carriers. Concerning instead the superconducting phase, hydrogen-intercalated $TiSe_2$ does not show the dome behavior in doping that characterizes other transition metal dichalcogenides, the superconducting critical temperature $T_c$ being finite and constant as the doping is varied.

A structure similar to the ion-gated $TiSe_2$ was observed by scanning transmission microscope measurements of the closely related $1T$-$TaS_2$, in which the incommensurate charge-density-wave state is found even at ambient conditions [33]. Self-organization results from the repulsive interactions between domain walls, which are associated with higher-order terms in the free energy [34]. Therefore, the incommensurate charge-density-wave state will form a matrix, fragmenting the commensurate charge-density-wave state into domains with fixed area, as needed for the occurrence of the Little–Parks effect. As shown in Ref. [34], incommensurate charge-density-wave dynamic phase fluctuations, i.e., the phonon modes of the incommensurate charge-density wave, can occur within the domain walls. It is plausible that these incommensurate charge-density-wave phonons may trigger superconducting pairing and localize the resulting Cooper pairs in one-dimensional regions of the two-dimensional system. Another intriguing result of Ref. [34] is that the point-contact conductance spectra measured at each carrier density is characterized by the presence of a zero-bias conductance peak in the superconducting state. Zero-bias conductance peaks are observed in a variety of seemingly unconventional superconductors and are interpreted as the consequence of Andreev reflection promoted by a Cooper pairing potential with an internal phase shift of the superconductivity order parameter [35].

The coexistence of commensurate and incommensurate charge-density waves was first observed by X-ray measurements on $TiSe_2$ at pressures close to those at which superconductivity was expected to occur [36]. Incommensurate charge-density-wave domain walls with a periodicity along the $c$ axis of approximately $300\,nm$ were observed, which is similar to the length scale determined in Ref. [29].

X-ray diffraction measurements on $Cu_x$-$TiSe_2$ [37] highlighted an incommensurate charge-density wave, occurring at an intercalant concentration which coincided with the onset of superconductivity. This result seems to agree with an increasing number of

experiments pointing to the relevance of incommensurate charge-density waves for the occurrence of superconductivity in transition metal dichalcogenides [36,38–40], although the role of crystallographic disorder could not be ruled out. In addition, the authors of Ref. [37] showed that the charge-density waves do not terminate near (or inside) the superconducting dome, and rather, they survive up to an intercalant concentration much larger than previously thought.

It is worth noting that in 1T-TaS$_2$, the commensurate charge-density wave is destabilized by pressure or Li ion intercalation, whereas the incommensurate charge-density wave survives and coexists with superconductivity [39]. Likewise, in 1T-TaS$_{2-x}$Se$_x$, superconductivity is sandwiched between two regions of commensurate charge-density waves as $x$ is varied, and  again, it only coexists with incommensurate charge-density waves [38]. In the case of 2H-TaSe$_2$, the superconducting transition temperature can be raised to 2 K by irradiating the sample, thereby disrupting the commensuration of the charge-density wave and introducing disorder [40]. Interestingly, the incommensuration of the charge-density wave can be distinguished only at about the same intercalation content at which superconductivity sets in. This is similar to what is observed in TiSe$_2$ under pressure, where incommensurate fluctuations were highlighted above the superconducting dome [36].

Likely, the peculiarities of the phase diagram of Cu$_x$-TiSe$_2$ are the consequence of Cu intercalants electron doping the Ti-3$d$ conduction band [41], thereby suppressing excitonic correlations. Many studies led to the conclusion that both electron–phonon coupling and electron–hole coupling drive the charge-density-wave transition in pure 1T-TiSe$_2$ [42–51]. Electron doping selectively weakens the excitonic contribution to the charge-density wave by shifting the chemical potential into the conduction band, thereby enhancing screening effects, while leaving the electron–phonon interaction substantially unaffected. This interpretation provides a natural framework to interpret the peculiarities of the phase diagram of Cu$_x$-TiSe$_2$: both excitonic and electron–phonon interactions drive the charge-density-wave transition in the low-intercalation region of the phase diagram, while only electron–phonon coupling plays a role in the high-intercalation region.

Scanning tunneling microscopy measurements [32], meant to investigate the interplay between charge-density waves and superconductivity in 1T-Cu$_x$TiSe$_2$, confirmed that the implication of Cu atoms in the observed alterations of charge-density waves is challenging, because Cu atoms and charge-density waves cannot be simultaneously probed. Indeed, Cu atoms are only resolved at negative bias voltages, less than $-800$ mV [52,53], while charge-density-wave contrast is achieved at lower bias voltages within a few hundred meV of the charge-density-wave gap [54]. To obtain the alignment of images taken at such different biases, fingerprints of atomic defects visible at all biases, in particular intercalated Ti, were used [54,55]. An interesting feature of the charge-density waves in Cu-intercalated TiSe$_2$ is the presence of an inhomogeneous electron background. Such an inhomogeneity is directly related to intercalated Cu atoms, which tend to cluster. Seemingly, Cu intercalation affects the long-range $2 \times 2$ commensurate charge-density wave observed in the $ab$ plane of pristine crystals in two different ways: it induces a sizable energy-dependent patchwork of charge-density-wave regions while promoting the formation of $\pi$-phase shift domain walls. The contrast inversion expected for a standard electron–hole symmetric charge-density wave [56] is missing.

Finally, the charge-density-wave pattern sheds new light on the properties of ion-liquid-gated TiSe$_2$ films [29], resulting in a spatially inhomogeneous carrier distribution [9,57,58]; the associated nonuniform potential landscape is expected to promote energy-dependent charge-density-wave patches, similar to those observed in Ref. [32].

The observed charge-density-wave pattern provides evidence that the charge-density-wave gap in 1T-Cu$_x$TiSe$_2$ opens below the Fermi level and moves to higher binding energies with increasing Cu content. Remarkably, the charge-density wave probed by scanning tunneling microscopy survives for Cu doping deep inside the superconducting dome, pointing at a possible coexistence of the two phases.

### 2.2. High-Critical Temperature Superconducting Cuprates

The phase diagram of high-critical temperature superconducting cuprates exhibits a variety of competing phases. Hereafter, we discuss the more common hole-doped cuprates. The undoped parent compound is an antiferromagnetic Mott insulator, but antiferromagnetism is rapidly suppressed with increasing doping, the Néel temperature vanishing for doping $p \geq 0.02$. The superconducting critical temperature $T_c$ increases and then decreases upon doping, giving rise to a characteristic dome-shaped curve $T_c(p)$ in the temperature vs. doping phase diagram (see Figure 2). The optimal doping corresponds to the highest $T_c$; samples with lower (higher) doping are said to be underdoped (overdoped). In the underdoped region of the phase diagram, the electron density of states at the Fermi energy is partially suppressed below the so-called pseudogap crossover temperature $T^*$ [59], with $T^*(p)$ decreasing with increasing doping and reaching $T_c$ around optimal doping. Polarized neutron scattering experiments suggest a time-reversal and inversion symmetry breaking below $T^*$ in $YBa_2Cu_3O_{6+\delta}$ (YBCO) [60–63] that might be mirrored in the symmetry of the superconducting state [64]. This symmetry breaking was subsequently observed in $La_{2-x}Sr_xCuO_4$ (LSCO) [65], $Bi_2Sr_2CaCuO_{8+\delta}$ (BSCCO) [66,67], and $HgBa_2CuO_{4+\delta}$ (HBCO) [68,69].

Under a strong magnetic field, the very topology of the Fermi surface seems to be strongly affected in the underdoped region of the phase diagram, where magnetotransport studies [70–73] suggest that a reconstruction of the Fermi surface takes place in YBCO in the doping range $0.08 \lesssim p \lesssim 0.16$. More recent work on YBCO also suggests that a sudden change in the carrier density occurs around the endpoint of the line $T^*(p)$, at $p \approx 0.19$ [74], which is in agreement with an early scenario based on optical conductivity computations [75], suggesting a restructuring of the Fermi surface with a sizable reduction of its volume at low doping. Under the the same conditions, a static charge ordering is observed by nuclear magnetic resonance [76] and by X-ray scattering [77], setting in at a temperature lower than $T_c(p)$. Above $T_c$, resonant X-ray scattering experiments [78–83] revealed the presence of dynamical charge-density waves, and nuclear magnetic resonance experiments [84] indicate that static short-range charge-density waves may be pinned by the existing disorder, even at low magnetic field. This rich phenomenology naturally suggests the interplay of various physical mechanisms and produced a wealth of different theoretical proposals. For instance, the pseudogapped state observed below $T^*(p)$ might be due to an exotic metallic state resulting from strong electron–electron correlations in the proximity of the Mott insulating phase [85], or it might be related to a missed quantum critical point, hidden underneath the superconducting dome and associated to some ordered state, such as circulating currents [86,87] or charge order [88–93].

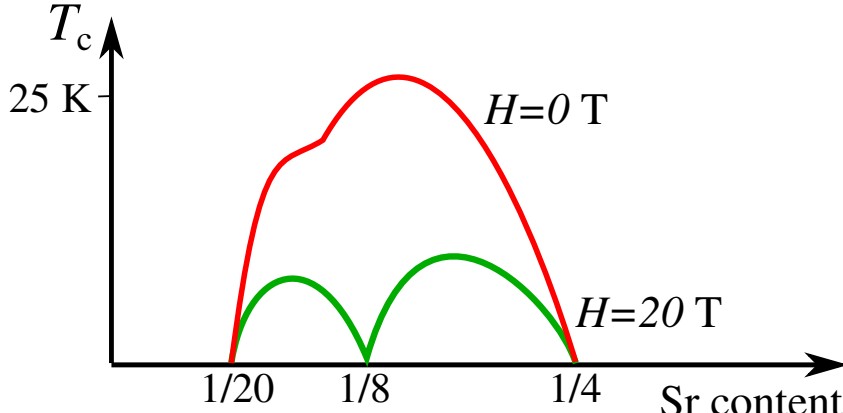

**Figure 2.** Superconducting critical temperature $T_c$ (defined as the occurrence of a zero-resistance state) of LSCO as a function of doping (Sr content) in the absence of magnetic field (red curve, labeled with $H = 0\,T$) and for a magnetic field $H = 20\,T$ (green curve). The presence of two distinct domes at high magnetic field was also observed in YBCO in Ref. [94]. The sketchy figure is adapted from Ref. [95], where $T_c$ curves at intermediate values of the magnetic field and other details can be found.

Some authors suggest that antiferromagnetic fluctuations can play a role [96], despite the fact that the endpoint of the antiferromagnetic critical line is located in the underdoped region outside the superconducting dome.

Other authors propose that the presence of Cooper pairs might account for the properties of the pseudogapped state. However, the existence of preformed Cooper pairs below $T^*$ has been questioned [97,98], and paraconductivity measurements in the pseudogapped state of LSCO [97,99,100] show that customary Cooper pair fluctuations occur over a wide temperature range above $T_c$.

In Ref. [95], a thorough survey of the resistance measurement under a strong magnetic field in LSCO thin films was undertaken. The samples covered a wide doping range, from heavily underdoped to heavily overdoped ($0.045 \leq x \leq 0.27$). The strong magnetic fields was used to tilt the balance between different phases across the whole doping range. The resulting phase diagram was in excellent agreement with a scenario in which, at low doping, disorder drives filamentary superconductivity inside an otherwise charge-ordered phase [101], which was conjectured to be a charge-density-wave phase, despite the fact that magnetotransport experiments cannot unambiguously identify the order parameter, and other scenarios involving different ordered states [85,102] could not be ruled out.

We remark that what appears as a single dome at zero and low $H$, for large enough $H$ splits into two domes that are centered, respectively, around Sr contents $x \approx 0.09$ and $x \approx 0.19$ (with a maximum at $x \approx 0.16$), similarly to what was previously observed in YBCO [94] and also inferred in Ref. [103]. The separation between the two domes is located at $x = \frac{1}{8}$. Two possible scenarios arise [95]: (i) commensuration effects may favor charge-density waves at $x = \frac{1}{8}$, producing a dip in the $T_c(x)$ dome, and a stronger magnetic field weakens superconductivity and deepens the dip, until the superconducting dome is split in two; (ii) increasing the magnetic field suppresses superconductivity, thereby favoring the competing charge-density-wave state, except near the endpoints $x_1$ and $x_2$ of the charge-density-wave dome hidden underneath the superconducting dome, where the strong charge fluctuations around the quantum critical points would enhance pairing [104], thereby strengthening superconductivity around $x_1 \approx 0.09$ and $x_2 \approx 0.16$, similarly to what happens, e.g., in many heavy-fermion metals, where superconductivity arises near a quantum critical point [105].

Other hints for the interplay between superconductivity and another ordered state in cuprates come from the observation of a two-stage transition in LSCO [106], spin susceptibility measurements in YBCO [107] and specific heat measurements in YBCO [108], all suggesting charge-density waves to be the competing phase.

It must be pointed out that cuprates are very complicated systems, the various families sharing common aspects but also significant differences, and a lot of confusion arises about their properties, especially when the discussion is about ordered states that may compete with superconductivity. Our work falls within a line of research in which cuprates of all families are on the verge of choosing between superconductivity and (some form of) charge order, and superconductivity finally takes over. Remnants of the tendency to charge order survive in the form of dynamical charge-density waves, as detected by RIXS [109], unless something is done, purposely or accidentally, to stabilize them by means of a non-thermal parameter, such as, e.g., Nd-codoping, or a magnetic field, that suppresses superconductivity, uncovering static charge order underneath the superconducting dome. We anticipate here that an intriguing scenario arises since the two competing phases must have very similar (free) energies, the competition being tilted under the action of rather weak non-thermal disturbances, so the system is not far (in parameter space) from a situation when the two phases are degenerate (see Section 3). Likely, also structural changes may tilt the balance between superconductivity and the charge-density wave as a side effect [110,111]. If the two phases coexist, one as the stable phase and the other as a metastable phase, disorder can promote the formation of domains with different realizations of charge order [103], and filamentary superconductivity may occur as a

topologically protected parasitic phase at the domain walls between two regions with different realizations of charge order (see Section 4).

### 3. The Ginzburg–Landau Model for the Competition of Two Phases

In this section, aiming at investigating the possible outcomes within a scenario of competing superconductivity and charge-density waves, we adopt a Ginzburg–Landau approach. The free energy describing a system with two competing order parameters is

$$F(\Delta, \varphi) = \frac{a}{2}\Delta^2 + \frac{b}{4}\Delta^4 + \frac{c}{2}\varphi^2 + \frac{d}{4}\varphi^4 + \frac{e}{2}\Delta^2\varphi^2 \qquad (1)$$

where $\Delta$ ($\varphi$) corresponds to the superconducting (charge-density-wave) order parameter, the last term setting the competition if $e > 0$. In the simplest realization, the symmetry of the two order parameters is $Z_2$ (Ising-like), but the generalization to $U(1)$ symmetry of the superconducting order parameter is straightforward, promoting $\Delta$ to a complex order parameter and interpreting $\Delta^2$ as $|\Delta|^2$ in Equation (1). At the mean-field level, anyway, in the absence of external fields coupled to the superconducting order parameter, one can always chose the phase of $\Delta$ to be zero. We also point out that the Ginzburg–Landau theory of a $d$-wave superconductor is essentially the same as that of a $s$-wave superconductor, so our theory applies to both. Time reversal symmetry is not broken in the simplest realization of our scenario, while of course, more elaborated versions can be worked out and endowed with more exotic behaviors. Our model was extensively studied by Imry [112] and, more recently, by Lee and collaborators in the Supplementary Materials of Ref. [113], in the context of nickel-based pnictides. Here, we only discuss the regimes that may be relevant to our forthcoming analysis, adopting the simplest possible description of the two competing phases and their interplay. Thus, for the sake of simplicity, in the following, we assume that $b$ and $d$ are nearly constant (i.e., weakly dependent on the external control parameters, such as the temperature, the magnetic field, or the carrier density), and we take $b = d = 1$, which amounts to rescaling $\Delta$ and $\varphi$. Within a mean-field description, one can easily find the various phases of the model minimizing the free energy in (1) with respect to $\Delta$ and $\varphi$ and identifying the conditions for the existence and stability of the various phases. The solutions that make the gradient of the free energy vanish (extremal points) are

(1)     $\Delta = \varphi = 0$,

(2)     $\begin{cases} \Delta^2 = -a, & a < 0, \\ \varphi = 0, \end{cases}$     (3)     $\begin{cases} \Delta = 0 \\ \varphi^2 = -c, & c < 0, \end{cases}$

(4)     $\begin{cases} \Delta^2 = \frac{a-ec}{e^2-1}, \\ \varphi^2 = \frac{c-ea}{e^2-1}. \end{cases}$

This last solution, with both order parameters different from zero, exists only if $a, c < 0$ and either

$$\begin{cases} 0 < e < 1, \\ a/e < c < ea, \end{cases} \quad \text{or} \quad \begin{cases} e > 1, \\ ea < c < a/e. \end{cases} \qquad (2)$$

By inserting those solutions in (1), one finds

$$F_1 = 0, \quad F_2 = -\frac{a^2}{4}, \quad F_3 = -\frac{c^2}{4}, \quad F_4 = \frac{a^2 + c^2 - 2ace}{4(e^2-1)}. \qquad (3)$$

The nature of various extremum points (minima, maxima, saddle points) is easily ascertained by inspection of the corresponding Hessian matrix.

The solution $\Delta = \varphi = 0$ describes the disordered phase. This is a minimum of the free energy when both $a$ and $c$ are positive, a saddle point if $a$ and $c$ have opposite signs, and a maximum when both $a$ and $c$ are negative. This means that the disordered phase is stable in the first quadrant of the $c$ vs. $a$ plane in parameter space. The phase with $\Delta \neq 0$ and

$\varphi = 0$, describing the superconducting phase, is a minimum for $a < 0$ and $c > ea$; the phase with $\Delta = 0$ and $\varphi \neq 0$, describing the charge-ordered phase, is a minimum for $c < 0$ and $c < a/e$. In the following, we shall call these two phases the pure phases to be contrasted with the mixed phase where both $\Delta$ and $\varphi$ are simultaneously nonzero.

To discuss the ordered phases of our model in more detail, let us consider separately the cases $e > 1$ and $0 < e < 1$. When $e > 1$, we find that $F_{2,3} < F_4$ for all $ea < c < a/e$, where the mixed solution with both nonzero order parameters exists. The half line $c = a$ in the third quadrant of the $c$ vs. $a$ plane represents the first-order transition line between the two pure phases, but the two pure phases can coexist (one as the stable phase, the other as a metastable phase) in the region of the third quadrant of the $c$ vs. $a$ plane encompassed by the two lines $c = ea$ and $c = a/e$, which represent the spinodal lines for the phases with $\Delta \neq 0$ and $\varphi \neq 0$, respectively (see Figure 3b).

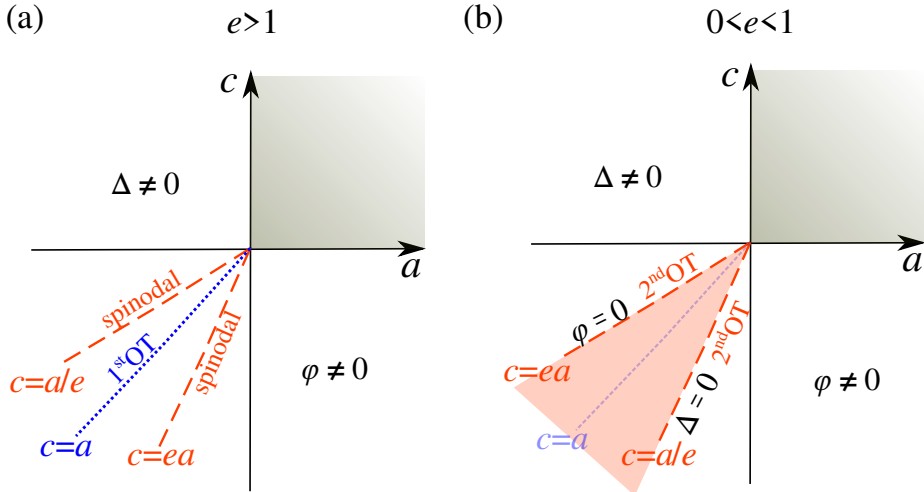

**Figure 3.** Phase diagram in the $c$ vs. $a$ plane for the simple model encoded in (1). (**a**) $e > 1$, the two ordered phases with $\Delta \neq 0$ (stable in the half-plane $a < 0$) and with $\varphi \neq 0$ (stable in the half-plane $c < 0$) are strongly competing. The half-line $c = a$ represents the first-order transition line between the two phases, but the two phases can coexist (one as the stable phase, the other as a metastable phase) in the whole third quadrant of the $c$ vs. $a$ plane, with the two half-lines $a = 0$, $c < 0$ and $a < 0$, $c = 0$, representing the spinodal lines for the phases with $\Delta \neq 0$ and $\varphi \neq 0$, respectively; (**b**) $0 < e < 1$, where instead there is a finite region (highlighted in orange) in which both $\Delta$ and $\varphi$ are simultaneously nonzero. The gray shaded regions in the first quadrant of both panels represent the disordered phase ($\Delta = \varphi = 0$).

Conversely, when $0 < e < 1$, we find that $F_4 < F_{2,3}$ in the whole region $a/e < c < ea$ of parameter space, where the mixed solution with both nonzero order parameters exists. In this latter case, the mixed phase joins continuously the pure phase with $\Delta \neq 0$ and $\varphi = 0$ along the half-line $c = ea$ in the third quadrant of the $c$ vs. $a$ plane, and the pure phase with $\Delta = 0$ and $\varphi \neq 0$ along the half-line $c = a/e$ in the third quadrant of the $c$ vs. $a$ plane (see Figure 3a). Thus, for positive but sufficiently small $e$, the competition between the two phases is not strong, and the first-order phase transition between the pure phases characterized by either of the two competing orders is circumvented by the occurrence of a mixed phase where both $\Delta$ and $\varphi$ are nonzero.

The origin of the phase diagrams in Figure 3 is a bicritical point when $e > 1$, and two second-order critical lines, $\{c = 0, a > 0\}$ and $\{a = 0, c > 0\}$ meet with a first-order line, $\{a = c, a, c < 0\}$, and a quadricritical point when $0 < e < 1$, and four second-order critical lines, $\{c = 0, a > 0\}$ $\{a = 0, c > 0\}$, $\{c = ea, a, c < 0\}$, and $\{c = a/e, a, c < 0\}$, meet. In this latter case, the mixed phase with both $\Delta$ and $\varphi$ different from zero is the least symmetric of all phases (both symmetries are broken).

In other words, for weak competition of the two order parameters, the two symmetries can be simultaneously broken, meaning that the two superconducting and charge-density-wave gaps are present at the same time and uniformly in the system. Conversely, in the strong competition regime, a (meta)stable phase with both nonzero order parameters cannot exist. The system exhibits a first-order phase transition between a pure charge-density-wave phase and a pure superconducting phase. In the region of coexistence of the two minima, a metastable state can nucleate, with a lifetime that decreases exponentially with the height of the potential barrier and goes to zero in the thermodynamic limit, according to Arrhenius law. In absence of any other external source that can stabilize it, the observation of a metastable state is known to be quite difficult. However, disorder can act as a source of nucleation. A dirty system presenting a strong competition between the two phases would hence provide a good playground to observe a phase-separated system that might support, e.g., the occurrence of a filamentary superconducting structure.

We note, on passing, that the limiting case $e = 1$ is peculiar, in that it makes the model supersymmetric along the half-line $c = a$ in the third quadrant of the $c$ vs. $a$ plane. Along this line, the pure phases with $\Delta \neq 0$ and $\varphi \neq 0$ are degenerate to all the phases with the same magnitude of $\sqrt{\Delta^2 + \varphi^2}$ (this phase might be called supersolid).

## 4. Competition between Superconductivity and Charge-Density Waves as a Mechanism Promoting Filamentary Superconductivity

Let us now discuss the phase diagram proposed for cuprates in Ref. [95], taking the various characteristic temperature scales appearing in the resistance vs. temperature curves measured in LSCO at various magnetic fields (see Figure 4) as proxies of the various phases. At high magnetic fields, the resistance curves are characterized by a minimum at a temperature $T_{MIN}(H)$ that decreases with decreasing $H$ (red and yellow curves in panel **b** show the corresponding resistances). This temperature scale was proposed to be the proxy of charge-density-wave ordering. At zero or low magnetic field, the resistance curves decrease monotonically with decreasing the temperature. We take the temperature of the inflection point $T_{INF}(H)$ as the proxy for the onset of superconducting fluctuations, which decreases with increasing $H$ and signals the precursor of superconductivity (see also the purple curve in panel b). The two curves $T_{MIN}(H)$ and $T_{INF}(H)$ meet, with a vertical tangent, at a magnetic field $H_c^*$ that marks at zero temperature the quantum critical point QCP* for the magnetic-field-driven transition between superconductivity and charge-density wave. The corresponding resistance, plotted in blue, shows in fact a horizontal tangent signaling the (avoided) quantum critical point, and then, it drops to zero at some lower temperature $T_c$. According to the scenario discussed in Section 3, cuprates appear to be in a regime of strong competition between the two phases ($e > 1$), and the transition is of first order. The first-order transition line is very steep (nearly vertical) in the temperature vs. magnetic field plane (see Figure 4).

Surprisingly, however, the resistance curves develop a remarkable non-monotonicity in a window of magnetic fields $H_c^* < H < H_c$: with decreasing the temperature, after reaching the minimum at $T = T_{MIN}(H)$, the resistance curves reach a maximum at $T = T_{MAX}(H)$ and then decrease, as sketched by the yellow curve of panel b, signaling the appearance of a low-temperature superconducting state, surviving inside the charge-density-wave state up to a magnetic field $H_c$ larger than $H_c^*$ (see Figure 4).

We can compare the above phenomenology with the results of Section 3 in the case $e > 1$, which crudely captures the competition among superconductivity and charge-density waves. The two phases strongly compete, which is presumably because the corresponding condensates gain energy from the same states in momentum space (that can condense either in the particle–particle or in the particle–hole channel). Once the parameters $a$ and $c$ of Equation (1) are assumed to depend on the temperature $T$ and on the magnetic field $H$, the various lines in Figure 3a translate into lines in the $T$ vs. $H$ plane. We suggest that the proxy for superconductivity, $T_{INF}(H)$ in Figure 4 corresponds to the mean-field critical line described by the implicit equation $a(T, H) = 0$, with $c(T, H) > 0$,

of Figure 3a; see Figure 5. The proxy for CDW, $T_{MIN}(H)$ in Figure 4 corresponds to the mean-field critical line described by the implicit equation $c(T, H) = 0$, with $a(T, H) > 0$, of Figure 3a; see Figure 5. The two ordered phases meet with the unbroken symmetry phase, $\Delta = \varphi = 0$, at the bicritical point $a(T, H) = c(T, H) = 0$, which is marked with the symbol BP in Figure 5. The bicritical point is the endpoint of a first-order critical line, $a(T, H) = c(T, H)$, with $a(T, H), c(T, H) < 0$, which in cuprates seems to be nearly independent of the temperature, terminates at $T = 0$ in correspondence of the magnetic field $H = H_c^*$, and separates the superconducting and charge-density-wave phases. Fluctuations beyond the mean field shrink the ordered regions of the phase diagram, bringing the superconducting transition from $T_{INF}(H)$ down to $T_c(H)$ and the charge-density-wave transition from $T_{MIN}(H)$ down to a corresponding transition line $T_{CDW}(H)$ (not marked in Figure 4).

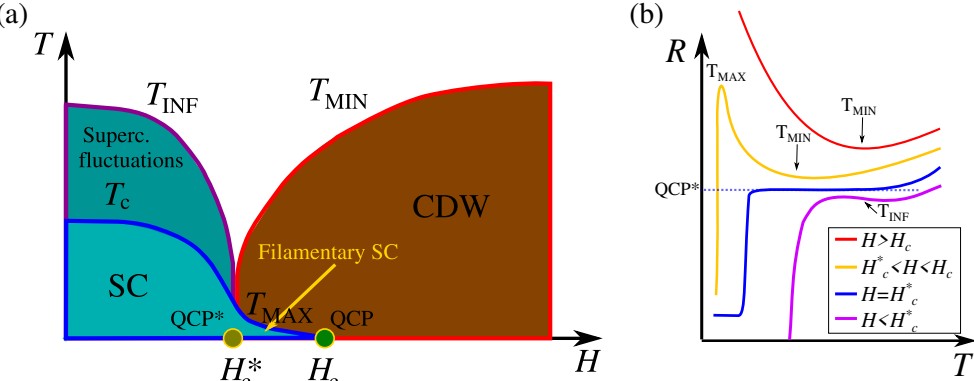

**Figure 4.** (**a**) Phase diagram of the temperature $T$ vs. magnetic filed $H$ proposed for LSCO based on the characteristic temperature scales identified on the resistance curves measured at various magnetic fields: $T_{MIN}(H)$, $T_{MAX}(H)$, and $T_{INF}(H)$ are, respectively, the temperature of the minimum, of the maximum, and of the inflection point found along the resistance curves, while $T_c$ is the temperature of the zero-resistance state. At zero temperature, two characteristic values of the magnetic field are found, $H_c^*$ and $H_c$, which represent the missed quantum critical point QCP* (see text) and the quantum critical point for the disappearance of superconductivity, respectively. Filamentary superconductivity is suggested to occur for $H_c^* < H < H_c$. Figure adapted from Ref. [95], where all the details can be found. (**b**) Cartoon plot of the resistance measured for various magnetic fields in Ref. [95] and sketched in Ref. [101].

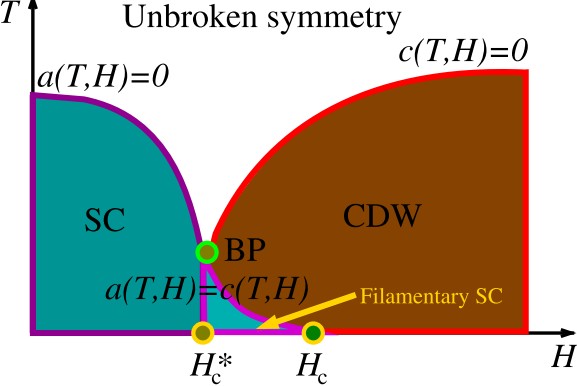

**Figure 5.** Phase diagram resulting from the comparison of the results of the results of Section 3 and the related mean-field phase diagram, Figure 3a, with the phase diagram conjectured in Ref. [95] and displayed in Figure 4. The label BP marks the bicritical point where the first-order transition line that separates superconductivity and charge-density waves meets the two critical lines separating superconductivity and charge-density waves from the unbroken symmetry phase. The filamentary superconducting state is highlighted by a lighter shade of cyan.

The scenario discussed so far does not yet allow for the occurrence of filamentary superconductivity. However, in Refs. [95,101], it was argued that disorder, which is not taken into account in Equation (1), might play a crucial role in promoting the onset of filamentary superconductivity, which was thereby conjectured to occur in cuprates for fields $H_c^* < H < H_c$ as a parasitic phase at sufficiently low temperature; see Figures 4 and 5.

Specifically, disorder is expected to mainly couple to charge-density waves, promoting the fragmentation of the charge-density-wave state into domains with different realizations of charge order (in our crude model, regions with $\varphi > 0$ and regions with $\varphi < 0$). In the region of the phase diagram where superconductivity exists as a metastable phase (the region between the half-line $c = a$ and the half-line $c = ea$ in the third quadrant of Figure 3a) can be promoted to a locally stable phase within the domain walls separating the domains with different realizations of charge order, possibly giving rise to a filamentary superconducting state, as highlighted by a lighter shade of cyan in Figure 5.

To understand why and how the competition between charge-density waves and superconductivity in the presence of disorder can trigger the formation of superconducting filaments, we can follow the line of reasoning of Ref. [101]. The model introduced there belongs to a class of models in which charge-density waves and superconductivity are two manifestations of a (missed) wider symmetry. In our Ginzburg–Landau model, as shown in Equation (1), this wider symmetry is achieved for $e = 1$ and $a = c$, as discussed at the end of Section 3, and it is only approximate for $e \neq 1$ and/or $a \neq c$. The simplest realization of the wider symmetry is $SO(3)$, and it is decomposed into $Z_2 \times U(1)$, where $Z_2$ is an Ising-like symmetry describing two different realizations of a commensurate charge-density wave, e.g., two possible choices for the location of the higher density site in the lattice, and $U(1)$ is the standard symmetry of the complex order parameter of a superconductor. The two competing phases were thus encoded in a three-dimensional vector $\vec{\Psi} = (\text{Re}\,\Delta, \text{Im}\,\Delta, \varphi)$, which is suitably normalized so that its tip can reach any point of a sphere (see Figure 6). The two different realizations of the charge-density waves (named A-CDW and B-CDW in Figure 6) correspond to the tip of $\vec{\Psi}$ located at the north or south pole of the sphere, respectively, whereas superconductivity is represented by the tip of $\vec{\Psi}$ reaching any point along the equator of the sphere, as appropriate to a phase that spontaneously breaks $U(1)$ symmetry. This description is particularly suitable when the statistical mechanical model describing the system enjoys approximate $SO(3)$ symmetry in the space spanned by $\vec{\Psi}$. When disorder promotes the fragmentation of the charge-density wave into neighboring domains hosting the two different realizations of the charge-density waves, the vector $\vec{\Psi}$ must gradually rotate from the north to the south pole of the sphere. In doing so, it needs to cross the equator, hence forming an in-plane domain wall in the form of superconducting filaments. If the filaments form a percolating network endowed with superconducting phase coherence (superfluid stiffness), the resulting filamentary superconducting phase can be seen as a topologically protected parasitic phase promoted by disorder [101].

We point out that when discussing the physics of cuprates, we always refer to a single copper–oxygen plane, putting aside the problem of phase coherence between the planes, that eventually results in three-dimensional (bulk) superconductivity. The occurrence of filamentary superconductivity may somewhat decouple the copper–oxygen planes, but further work is needed to investigate its connection with the suppression of three-dimensional superconductivity in favor of two-dimensional superconductivity [114], which is beyond the scope of the present article. It must be borne in mind that a filamentary superconducting state on a single copper–oxygen plane is somewhat one-dimensional-like. Still, in order to have a finite superfluid stiffness, the superconducting cluster must eventually be connected in a (very loose) two-dimensional structure.

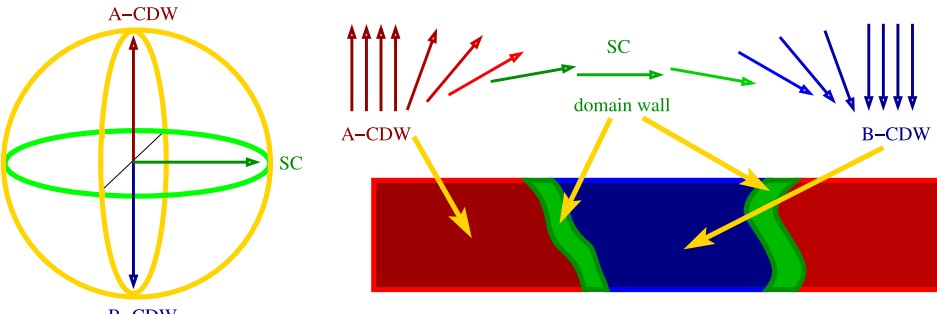

**Figure 6.** Sketch of the mechanism leading to the occurrence of filamentary superconductivity as a topologically protected parasitic phase. The three-dimensional vector $\vec{\Psi} = (\text{Re}\,\Delta, \text{Im}\,\Delta, \varphi)$, is normalized so that its tip can reach any point of a sphere (to the left in the figure). The two different realizations of the the charge-density waves (named A-CDW and B-CDW) correspond to the tip of $\vec{\Psi}$ located at the north or south pole of the sphere, respectively, whereas superconductivity is represented by the tip of $\vec{\Psi}$ reaching any point along the equator of the sphere. When the vector $\vec{\Psi}$ rotates within a domain wall (to the right in the figure) separating the two realizations of charge order (red and blue areas), it must necessarily pass through the equator so that the domain wall can support a superconducting filament (depicted in green). Figure adapted from Ref. [101].

## 5. Conclusions

To conclude, we proposed that a competition mechanism among superconductivity and charge-density waves can trigger the formation of filamentary superconductivity. Experimental evidence in both cuprates and some transition metal dichalcogenides indicates a coexistence of the two phases in suitable regions of their phase diagram. In both cases, this competition needs an external control parameter in order to be triggered. In the case of some transition metal dichalcogenides, the pristine material has a commensurate charge-density-wave ground state, which can be destroyed by chemical doping, pressure, or ion-gating techniques, to drive the appearance of superconductivity. On the other hand, the superconducting phase of hole-doped cuprates can be replaced by the charge-density-wave phase through the application of a magnetic field. Whereas the two materials are indeed very different in many aspects, we proposed to study the superconducting vs. charge-density-wave competition by means of a very essential model within the Ginzburg–Landau approach, where the two phases are encoded in the parameters $\varphi$ and $\Delta$ representing, respectively, the charge-density-wave and superconducting order parameters. We found that for weak competition, i.e., $0 < e < 1$ and $a/e < c < ea$, with $a, c < 0$, $\varphi$ and $\Delta$ are simultaneously and uniformly nonzero, i.e., the two order parameters coexist (Figure 3b). In the case of strong competition, $e > 1$, while there is no phase with both order parameters different from zero, the region of the phase diagram with $ea < c < a/$ and $a, c < 0$ hosts the two phases as local minima of the free energy (one stable, the other metastable) with the first-order transition line between the two pure phases, $a = c$, with $a, c < 0$, embedded in the coexistence region delimited by the two spinodal lines of Figure 3a.

In some transition metal dichalcogenides, the competition with an incommensurate charge-density wave may be weak, possibly allowing for a phase with both order parameters different from zero. The case of strong competition, in the presence of disorder, may instead be relevant for cuprates [95,101], in which a very fragile superconducting state was suggested to be responsible for an avoided quantum critical point and the peculiar behavior of the magnetoresistance curves (see Figure 4).

As effective as it was, the model presented in Ref. [101] was missing a crucial part of the discussion, i.e., the temperature dependence. One possible way to solve the model at finite temperature is to use Monte Carlo simulations to construct a $T$ vs $\alpha$ phase diagram, where $\alpha$ is an external (non-thermal) control parameter that allows switching between the two phases. Beyond the construction of a reliable phase diagram at finite temperature, such as the one sketched in Figure 4, a Monte Carlo study can also address the temperature behavior of some quantities that better characterize the nature of the various phases

and transitions. In particular, a two-dimensional superconductor undergoes a Berezinskii–Kosterlitz–Thouless phase transition instead of the phase transition with an order parameter, characterizing the three-dimensional superconductor, and the role of phase fluctuations is much more relevant than the usual Cooper pair fluctuations. In one-dimensional systems, instead, phase slips generated by either thermal or quantum fluctuations can break the long-range coherence. The behavior in temperature of the superfluid stiffness would hence present different peculiarities depending on whether the superconductor is bulk, two-dimensional, homogeneous, inhomogeneous, or filamentary. To assess the presence of a phase coherence, it is thus crucial to investigate the temperature behavior of the superfluid stiffness. Such a thorough Monte Carlo analysis is currently underway [115].

**Author Contributions:** Conceptualization, G.V. and S.C.; methodology, G.V. and S.C.; formal analysis, G.V. and S.C.; investigation, G.V. and S.C.; writing—original draft preparation, G.V. and S.C.; writing—review and editing, G.V. and S.C.; supervision, S.C.; funding acquisition, S.C. All authors have read and agreed to the published version of the manuscript.

**Funding:** We acknowledge financial support from the University of Rome Sapienza under the projects Ateneo 2020 (RM120172A8CC7CC7), Ateneo 2021 (RM12117A4A7FD11B), Ateneo 2022 (RM12218162CF9D05), from the Italian Ministero dell'Università e della Ricerca, under the Project PRIN 2017Z8TS5B, and from PNRR MUR project PE0000023-NQSTI.

**Data Availability Statement:** No new data were created or analyzed in this study.

**Acknowledgments:** We acknowledge stimulating discussions with M. Grilli, B . Leridon, J. Lorenzana, and I. Maccari.

**Conflicts of Interest:** The authors declare no conflict of interest. The funders had no role in the design of the study; in the collection, analyses, or interpretation of data; in the writing of the manuscript; or in the decision to publish the results.

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
