# Peer review of "Charge-Density Waves vs. Superconductivity: Some Results and Future Perspectives"

_condensedmatter, doi:10.3390/condmat8030054_

Round 1

Reviewer 1 Report

I find the scenario intriguing and probable that the superconductivity arises in a filamentary nature due to the competition between two phases as proposed by the authors of this article.   In fact, electronic phase separation is a topic that has been much studied in reference to other highly correlated systems such as the colossal magnetoresistance manganites. Here, the percolative phase separation of ferromagnetic conducting domains within an antiferromagnetic charge and orbital ordered lattice gives rise to the much studied property (see https://www.sciencedirect.com/science/article/pii/S0370157321000776 and reference therein). The authors might like to broaden out their discussion by referring to other electronic systems where such phase separation is of relevance.

I have some reservations about the degree to which the discussion emerges from the proposed Landau model rather than a speculative interpretation of other results already presented in the literature. However, if the authors can address these criticisms, and other comments given below, I believe this article is appropriate for publication in this journal.

I can’t comment much on section 2.1 as I am not familiar with the transition metal dichalcogenides. However, for section 2.2, I found it a little unclear as to the precise high-Tc superconducting cuprates the authors were refereeing to at certain given point. Systems based on La2CuO4 and those on YBCO have quite distinct behaviours in parts (and very different structures).  

For systems based on La2CuO4 there are various hints that the suppression of superconductivity is concurrent with the observation of (structural) phase separation (e.g. https://www.sciencedirect.com/science/article/pii/092145349190601T and https://www.nature.com/articles/s41598-022-18574-1 ). The authors may wish to support their arguments by referring to such experimental observations.

There is much discussion in the literature (e.g. Tranquada et) that in regions of the phase diagram of La2CuO4 where 3d superconductivity is suppressed, 2d superconductivity persists. Can the authors please comment how this picture fits with the proposed filamentary superconductivity (which is something more like 1 dimensional I suppose)?

For Section 3, could the authors please clarify the assumed symmetry (momentum, parity with respect to time reversal symmetry etc…) of their order parameters?  From Figure 5, if the order parameter for the charge density wave is related to electron density (wavefunction squared) it is unclear to me how the desired picture of a domain boundary in which the superconducting order parameter (wavefunction) can occur.

In the Landau potential, it is unclear to me that the scenario where both SC and CDW order parameters are non-zero must necessitate a phase separation. The authors assume the orderings are necessarily mutually exclusive. If so, could they please clarify this as I believe the authors have shown the coexistence of the order parameters, but not necessarily of distinct phases as stated in the conclusions.

In section 4, I am unclear to what extent the scenario of filamentary superconductivity arises form the authors present work.  This is maybe my greatest concern about the manuscript. I find the scenario very appealing but I feel there is quite a big leap here form the phase separation potentially implied by the Landau model in section 3 to the filamentary nature discussed in reference to some of the experimental results in the literature.

Reviewer 2 Report

The manuscript by Venditti and Caprara presents a study on how a filamentary superconducting state may arise from the competition of a superconducting phase and a charge density wave phase. The study is based on the development of a Ginzburg-Landau model and its comparison with existing experimental evidence on transition metal dichalcogenides and cuprates, which are extensively reviewed in the first part of the paper. Overall, I find the manuscript to be well written and both the analysis and the conclusion to be sound and well supported. I only have a few minor points for the authors to consider in their revised manuscript.

1) The current abstract, while accurate, is perhaps too succint (for example, nowhere does it state that the main results of the manuscript are obtained by means of the Ginzburg-Landau model). I suggest the authors to expand it in order to allow it to better capture the results and conclusions of the manuscript.

2) At lines 9-14, the authors list a plethora of experimentally-realized two-dimensional electron systems; similarly, at lines 27-29 the authors state that the quantum metallic behavior has been observed in several two-dimensional highly crystalline systems. In both cases, I think it would be appropriate to include relevant references for each entry, be it via individual studies or literature reviews.

3) To allow readers to better visualize the literature results reviewed in section 2.1, I suggest the authors to include an additional figure showing the sketched phase diagram of doped/pressurized transition metal dichalcogenides, similar to the current figure 1 showing the case of cuprates. A sketch of the commensurate CDW phase embedded in the incommensurate CDW network would also be quite helpful.

4) In section 2.1, most of the discussion is carried out concerning gated TiSe2 and copper-doped TiSe2. However, similar behaviors have also been reported in the case of Li-doped TiSe2 (Liao et al., Nat. Commun. 2021, doi:10.1038/s41467-021-25671-8) and H-doped TiSe2 (Piatti et al., arXiv:2205.12951, doi:10.48550/arXiv.2205.12951). Specifically, the Little-Parks effect has been reported in LixTiSe2, whereas the possible coexistence of CDW and superconductivity in the phase diagram of electron-doped TiSe2 was also suggested by transport measurements in LixTiSe2 and more recently by both transport and NMR measurements in HxTiSe2.

5) In section 3, it is clear that in the case of weak competition the Ginzburg-Landau model predicts that system can develop a mixed phase where both order parameters are non-zero. On the other hand, to me it is not intuitively clear what occurs in the case of strong competition, where "two pure phases can coexist (one as the stable phase, the other as a metastable phase)". Could the authors explain more diffusively how, in practice, does the stable+metastable phase coexistence differ from that occurring in the mixed phase? Does it involve phase separation?

6) Figure 3 reports the magnetic field dependence of several temperature scales obtained from the resistivity curves. In my opinion it would be helpful to add a panel with one-two typical resistivity curves where said temperatures are explicitly highlighted.

Concerning the English language, I only have a few very minor suggestions:

- At line 415, I feel that the use of the adjective "juicy" is too informal. I suggest the authors to substitute it with a more appropriate term (such as significant, interesting, etc.).
- Lines 229-230: it should be "strong magnetic field" singular, not "fields"  plural.
- Line 294: it should be "bicritical", not "bicritcal".
- Line 296: it should be "quadricritical", not "quadricritcal".
- Line 308: it should be "various magnetic fields" plural, not "field" singular.

Round 2

Reviewer 1 Report

Authors are to be commended for the thorough replies written to my comments and significant revisions made to the manuscript.

As well as improving key parts of their discussion, which has helped clarify some misunderstanding I had about their work, they have also softened the emphasis on the extent to which their work supports a scenario of filamentary superconductivity.

I am happy to recommend it for publication without further revision.

Reviewer 2 Report

The authors have properly and extensively addressed all my concerns, and I therefore consider the manuscript suitable for publication.